# GRAPH NEURAL NETWORKS AS MULTI-VIEW LEARNING

## ABSTRACT

Graph Neural Networks (GNNs) have demonstrated powerful representation capability in semi-supervised node classification. In this task, there are often three types of information – graph structure, node features, and node labels. Existing GNNs usually leverage both node features and graph structure by feature transformation and aggregation, following end-to-end training via node labels. In this paper, we change our perspective by considering these three types of information as three views of nodes. This perspective motivates us to design a new GNN framework as multi-view learning which enables alternating optimization training instead of end-to-end training, resulting in significantly improved computation and memory efficiency. Extensive experiments with different settings demonstrate the effectiveness and efficiency of the proposed method.

## 1 INTRODUCTION

Graph is a fundamental data structure that denotes pairwise relationships between entities in a wide variety of domains (Wu et al., 2019b; Ma & Tang, 2021). Semi-supervised node classification is one of the most crucial tasks on graphs. Given graph structure, node features, and labels on a part of nodes, this task aims to predict labels of the remaining nodes. In recent years, Graph Neural Networks (GNNs) have proven to be powerful in semi-supervised node classification (Gilmer et al., 2017; Kipf & Welling, 2016; Velickovic et al., 2017). Existing GNN models provide different architectures to leverage both graph structure and node features. Coupled GNNs, such as GCN (Kipf & Welling, 2016) and GAT (Velickovic et al., 2017), couple feature transformation and propagation to combine node feature and graph structure in each layer. Decoupled GNNs, such as APPNP (Klicpera et al., 2018), first transform node features and then propagate the transformed features with graph structure for multiple steps. Meanwhile, there are GNN models such as Graph-MLP (Hu et al., 2021) that extract graph structure as regularization when integrating with node features. Nevertheless, the majority of aforementioned GNNs utilize node labels via the loss function for end-to-end training.

In essence, existing GNNs have exploited three types of information to facilitate semi-supervised node classification. This understanding motivates us to change our perspective by considering these three types of information as three views of nodes. Then we can treat the design of GNN models as multi-view learning. The advantages of this new perspective are multi-fold. First, we can follow key steps in multi-view learning methods to design GNNs by investigating (1) how to capture node information from each view and (2) how to fuse information from three views. Such superiority offers us tremendous flexibility to develop new GNN models. Second, multi-view learning has been extensively studied (Xu et al., 2013) and there is a large body of literature that can open new doors for us to advance GNN models.

To demonstrate the potential of this new perspective, following a traditional multi-view learning method (Xia et al., 2010), we introduce a shared latent variable to explore these three views simultaneously in a multi-view learning framework for graph neural networks (MULTIVIEW4GNN). The proposed framework MULTIVIEW4GNN can be conveniently optimized in an alternating way, which remarkably alleviates the computational and memory inefficiency issues of the end-to-end GNNs. Extensive experiments under different settings demonstrate that MULTIVIEW4GNN can achieve comparable or even better performance than the end-to-end trained GNNs especially when the labeling rate is low, but it has significantly better computation and memory efficiency.

## 2 THE PROPOSED FRAMEWORK

We use bold upper-case letters such as $\mathbf{X}$ to denote matrices. $\mathbf{X}_i$ denotes its $i$-th row and $\mathbf{X}_{ij}$ indicates the $i$-th row and $j$-th column element. We use bold lower-case letters such as $\mathbf{x}$ to denote vectors. The Frobenius norm and trace of a matrix $\mathbf{X}$ are defined as $\|\mathbf{X}\|_F = \sqrt{\sum_{ij} \mathbf{X}_{ij}^2}$ and $tr(\mathbf{X}) = \sum_i \mathbf{X}_{ii}$. Let $\mathcal{G} = (\mathcal{V}, \mathcal{E})$ be a graph, where $\mathcal{V}$ is the node set and $\mathcal{E}$ is the edge set. $\mathcal{N}_i$ denotes the neighborhood node set for node $v_i$. The graph can be represented by an adjacency matrix $\mathbf{A} \in \mathbb{R}^{n \times n}$, where $\mathbf{A}_{ij} > 0$ indices that there exists an edge between nodes $v_i$ and $v_j$ in $\mathcal{G}$, or otherwise $\mathbf{A}_{ij} = 0$. Let $\mathbf{D} = diag(d_1, d_2, \ldots, d_n)$ be the degree matrix, where $d_i = \sum_j \mathbf{A}_{ij}$ is the degree of node $v_i$. The graph Laplacian matrix is defined as $\mathbf{L} = \mathbf{D} - \mathbf{A}$. We define the normalized adjacency matrix as $\tilde{\mathbf{A}} = \mathbf{D}^{-\frac{1}{2}} \mathbf{A} \mathbf{D}^{-\frac{1}{2}}$ and the normalized Laplacian matrix as $\tilde{\mathbf{L}} = \mathbf{I} - \tilde{\mathbf{A}}$. Furthermore, suppose that each node is associated with a $d$-dimensional feature $\mathbf{x}$ and we use $\mathbf{X} = [\mathbf{x}_1, \mathbf{x}_2, \ldots, \mathbf{x}_n]^\top \in \mathbb{R}^{n \times d}$ to denote the feature matrix.

In this work, we focus on the node classification task on graphs. Given a graph $\mathcal{G} = \{\mathbf{A}, \mathbf{X}\}$ and a partial set of labels $\mathcal{Y}_L = \{\mathbf{y}_1, \mathbf{y}_2, \ldots, \mathbf{y}_l\}$ for node set $\mathcal{V}_L = \{v_1, v_2, \ldots, v_l\}$, where $\mathbf{y}_i \in \mathbb{R}^C$ is a one-hot vector with $C$ classes, our goal is to predict labels of unlabeled nodes. The labels of graph $\mathcal{G}$ can also be represented as a label matrix $\mathbf{Y} \in \mathbb{R}^{n \times C}$, where $\mathbf{Y}_i = \mathbf{y}_i$ if $v_i \in \mathbf{V}_L$ and $\mathbf{Y}_i = \mathbf{0}$ if $v_i \in \mathbf{V}_U$. The subscript $U$ and $L$ denote the sets of unlabeled and labeled nodes, respectively.

### 2.1 MULTI-VIEW LEARNING FOR GNNS

For the node classification task, we take a new perspective that considers node feature $\mathbf{X}$, graph structure $\mathbf{A}$, and node label $\mathbf{Y}$ as three views for nodes, and model graph neural networks as multi-view learning. In particular, we need to jointly model each view and integrate three views. To achieve this goal, we introduce a latent variable $\mathbf{F}$ inspired by a traditional multi-view learning method (Xia et al., 2010). Then the loss function can be written as:

$$\underset{\mathbf{F}, \Theta}{\arg\min} \, \mathcal{L} = \lambda_1 \mathcal{D}_X(\mathbf{X}, \mathbf{F}) + \mathcal{D}_A(\mathbf{A}, \mathbf{F}) + \lambda_2 \mathcal{D}_Y(\mathbf{Y}_L, \mathbf{F}_L), \tag{1}$$

where $\mathbf{F}$ is the introduced latent variable shared by three views, $\mathcal{D}_X(\cdot, \cdot)$, $\mathcal{D}_A(\cdot, \cdot)$ and $\mathcal{D}_Y(\cdot, \cdot)$ are functions to explore node feature, graph structure, and node label, respectively. These functions can contain parameters which we denote them as $\Theta$. Hyper-parameters $\lambda_1$ and $\lambda_2$ are introduced to balance the contributions from these three views. One major advantage of the multi-view learning perspective is that it enables immense flexibility to design GNN models. Specifically, based on (1), there are numerous designs for $\mathcal{D}_X(\cdot, \cdot)$, $\mathcal{D}_A(\cdot, \cdot)$ and $\mathcal{D}_Y(\cdot, \cdot)$. Examples are shown below:

- $\mathcal{D}_X$ is to map node features $\mathbf{X}$ to $\mathbf{F}$. In reality, we can first transform $\mathbf{X}$ before mapping. Thus, feature transformation methods can be applied including traditional methods such as PCA (Collins et al., 2001; Shen, 2009) and SVD (Godunov et al., 2021), and deep methods such as, MLP and self-attention (Vaswani et al., 2017). We also have various choices of the mapping functions such as Multi-Dimensional Scaling (MDS) (Hout et al., 2013) which preserves the pairwise distance between $\mathbf{X}$ and $\mathbf{F}$ and any distance measurements.
- $\mathcal{D}_A$ aims to impose constraints on the latent variable $\mathbf{F}$ with the graph structure. Traditional graph regularization techniques can also be employed. For instance, the Laplacian regularization (Yin et al., 2016) is to guide a node $i$'s feature $\mathbf{F}_i$ to be similar to its neighbors; Locally Linear Embedding (LLE) (Roweis & Saul, 2000) is to force the $\mathbf{F}_i$ be reconstructed from its neighbors. Moreover, modern deep graph learning methods can be applied, such as graph embedding methods (Perozzi et al., 2014; Grover & Leskovec, 2016) and Graph Contrastive Learning (Zhu et al., 2020; Hu et al., 2021), which implicitly encodes node similarity and dissimilarity.
- $\mathcal{D}_Y$ establishes the connection between the latent variable $\mathbf{F}_L$ and the ground truth node label $\mathbf{Y}_L$ for labeled nodes. It can be any classification loss function, such as the Mean Square Error and Cross Entropy Loss.

In this work, we set the dimensions of the latent variable $\mathbf{F}$ as $\mathbb{R}^{n \times C}$, which can be considered as a soft pseudo-label matrix. Then the following designs are chosen for these functions: (i) for $\mathcal{D}_X$, we use an MLP with parameter $\Theta$ to encode the features of node $i$ as $\text{MLP}(\mathbf{X}_i; \Theta)$, and then adopt the Euclidean distance to map $\mathbf{F}_i$ as $\|\text{MLP}(\mathbf{X}_i; \Theta) - \mathbf{F}_i\|_2^2$; (ii) for $\mathcal{D}_A$, Laplacian smoothness is imposed to constrain the distance between one node's pseudo labels $F_i$ and its neighbors as

$\sum_{(v_i,v_j) \in \mathcal{E}} \|\mathbf{F}_i/\sqrt{d_i} - \mathbf{F}_j/\sqrt{d_j}\|_2^2$; and (iii) for $\mathcal{D}_Y$, we adopt Mean Square Loss $\|\mathbf{F}_i - \mathbf{Y}_i\|_2^2$ to constraint the pseudo label of a labeled node close to its ground truth. These designs lead to our multi-view learning framework for graph neural networks (MULTIVIEW4GNN). Its loss function can be written in the matrix form as:

$$\mathcal{L} = \lambda_1 \underbrace{\|\text{MLP}(\mathbf{X}) - \mathbf{F}\|_F^2}_{\mathcal{D}_X} + \underbrace{\text{tr}(\mathbf{F}^\top \tilde{\mathbf{L}} \mathbf{F})}_{\mathcal{D}_A} + \lambda_2 \underbrace{\|\mathbf{F}_L - \mathbf{Y}_L\|_F^2}_{\mathcal{D}_Y}, \tag{2}$$

where the first term maps node features into the label space, the second term indicates the pseudo labels should be smooth over the graph, and the last term constrains that the pseudo labels should be close to the ground-truth labels for labeled nodes.

**Remark.** There are recent works (Zhu et al., 2021; Ma et al., 2021; Yang et al., 2021) that aim to provide a unified optimization framework for understanding the message passing mechanism of different GNNs and designing new graph filter layers. However, they only focus on the forward process without taking the backward learning process into consideration, and they still follows the existing GNN architecture with end-to-end training. In this work, we do not aim to understand the message passing and design new GNN layers based on existing architectures. Instead, MULTIVIEW4GNN is a new graph deep learning framework as multi-view learning.

## 2.2 An Alternating Optimization Method for Multiview4GNN

It is difficult to find the optimal solution of the loss function (2) for both $\mathbf{F}$ and $\Theta$ simultaneously due to the coupling between the latent variable $\mathbf{F}$ and model parameters $\Theta$. The alternating optimization (Bezdek & Hathaway, 2002) based iterative algorithm can be a natural solution for this challenge. Specifically, for each iteration, we first fix the model parameters $\Theta$ and update the shared latent variable $\mathbf{F}$ on all three views. Then, we fix $\mathbf{F}$ and update the parameters $\Theta$, which is effective in exploring the complementary characteristics of the three views. These two steps alternate until convergence. Next, we show the alternating optimization algorithm in detail.

**Update F.** Fixing MLP, we can minimize $\mathcal{L}$ with respect to the latent variable $\mathbf{F}$ using the gradient descent method. The gradient of $\mathcal{L}$ with respect to $\mathbf{F}$ (i.e., $\mathbf{F}_U$ and $\mathbf{F}_L$) is

$$\frac{\partial \mathcal{L}}{\partial \mathbf{F_L}} = 2\Big(\lambda_1(\mathbf{F}_L - \text{MLP}(\mathbf{X}_L)) + (\tilde{\mathbf{L}}\mathbf{F})_L + \lambda_2(\mathbf{F}_L - \mathbf{Y}_L)\Big), \tag{3}$$

$$\frac{\partial \mathcal{L}}{\partial \mathbf{F_U}} = 2\Big(\lambda_1(\mathbf{F}_U - \text{MLP}(\mathbf{X}_U)) + (\tilde{\mathbf{L}}\mathbf{F})_U\Big). \tag{4}$$

The gradient descent update of $\mathbf{F}$ with step sizes $\eta_L$ and $\eta_U$ is:

$$\begin{aligned} \mathbf{F}_L^{k+1} &= \mathbf{F}_L^k - 2\eta_L\Big(\lambda_1(\mathbf{F}_L^k - \text{MLP}(\mathbf{X}_L)) + (\tilde{\mathbf{L}}\mathbf{F}^k)_L + \lambda_2(\mathbf{F}_L^k - \mathbf{Y}_L)\Big) \\ &= 2\eta_L\Big((\tilde{\mathbf{A}}\mathbf{F}^k)_L + \lambda_1\text{MLP}(\mathbf{X}_L) + \lambda_2\mathbf{Y}_L\Big) + \Big(1 - 2\eta_L(\lambda_1 + \lambda_2 + 1)\Big)\mathbf{F}_L^k, \end{aligned} \tag{5}$$

$$\begin{aligned} \mathbf{F}_U^{k+1} &= \mathbf{F}_U^k - 2\eta_U\Big(\lambda_1(\mathbf{F}_U^k - \text{MLP}(\mathbf{X}_U)) + (\tilde{\mathbf{L}}\mathbf{F}^k)_U\Big) \\ &= 2\eta_U\Big((\tilde{\mathbf{A}}\mathbf{F}^k)_U + \lambda_1\text{MLP}(\mathbf{X}_U)\Big) + \Big(1 - 2\eta_U(\lambda_1 + 1)\Big)\mathbf{F}_U^k. \end{aligned} \tag{6}$$

According to the smoothness and strong convexity of the problem, we set $\eta_L = \eta_U = \frac{1}{2(\lambda_1 + \lambda_2 + 1)}$ to ensure the decrease of loss value $\mathcal{L}$ (Nesterov et al., 2018), and the update becomes:

$$\mathbf{F}_L^{k+1} = \frac{1}{\lambda_1 + \lambda_2 + 1}(\tilde{\mathbf{A}}\mathbf{F}^k)_L + \frac{\lambda_1}{\lambda_1 + \lambda_2 + 1}\text{MLP}(\mathbf{X}_L) + \frac{\lambda_2}{\lambda_1 + \lambda_2 + 1}\mathbf{Y}_L, \tag{7}$$

$$\mathbf{F}_U^{k+1} = \frac{1}{\lambda_1 + \lambda_2 + 1}(\tilde{\mathbf{A}}\mathbf{F}^k)_U + \frac{\lambda_1}{\lambda_1 + \lambda_2 + 1}\text{MLP}(\mathbf{X}_U) + \frac{\lambda_2}{\lambda_1 + \lambda_2 + 1}\mathbf{F}_U^k. \tag{8}$$

**Update $\Theta$.** Fixing $\mathbf{F}^{k+1}$, we can minimize the loss function $\mathcal{L}$ with respect to MLP parameters:

$$\arg\min_{\Theta} \|\text{MLP}(\mathbf{X}; \Theta) - \mathbf{F}^{k+1}\|_F^2, \tag{9}$$

which equals training the MLP with the soft pseudo labels via the Mean Square Error Loss. Besides, we can also apply the Cross-Entropy Loss, and the details are in Appendix A.

**Alternating Optimization and Scalability.** The multi-view learning perspective allows us to derive the alternating optimization solution which provides better flexibility in training than the end-to-end training in existing GNNs. The alternating optimization solution is a highly efficient and scalable training strategy, resulting in significantly improved computation and memory efficiency. Specifically, variable $\mathbf{F}$ and MLP model parameters $\Theta$ can be optimized separately. We can update $\mathbf{F}$ once and then train MLP multiple times. Meanwhile, there is no gradient backpropagation through the feature aggregation process so the aggregation steps do not need to store the activation and gradient values, which saves a significant amount of memory and computation. Moreover, due to the alternating updating scheme, we can use stochastic optimization to update $\mathbf{F}$ and train MLP by sampling the graph structure and node features. This can further improve the memory and computation efficiency as proved theoretically and empirically in stochastic optimization (Lan, 2020). In particular, only first-order neighbors are sampled in each update step of $\mathbf{F}$, which avoids the neighborhood explosion problem in training large-scale GNNs (Hamilton et al., 2017; Fey et al., 2021).

### 2.3 Understandings of Multiview4GNN

Another important advantage of alternating optimization is that it provides helpful insights to understand Multiview4GNN. In particular, based on the updating rules of $\mathbf{F}$ and $\Theta$, we can naturally draw the following understandings on Multiview4GNN.

**Understanding 1: Updating F is a feature-enhanced Label Propagation.** Label Propagation (LP) (Zhou et al., 2003) is a well-known graph semi-supervised learning method based on the label smoothing assumption that connected nodes are likely to have the same label. LP can be written in an iteration form: $\mathbf{F}^{(k+1)} = \alpha \tilde{\mathbf{A}} \mathbf{F}^{(k)} + (1 - \alpha) \mathbf{Y}$, where $\mathbf{F}^{(0)} = \mathbf{Y}$, $k$ is the propagation step, and $\alpha$ is a hyper-parameter. Comparing LP with our update rule for $\mathbf{F}$ in Eq. (7) and Eq. (8), we can find that MLP($\mathbf{X}$) is involved in the propagation where MLP($\mathbf{X}$) can be regarded as *labels* generated by node features. In other words, our update rule for $\mathbf{F}$ takes advantage of node features, graph structure and labels while LP only uses graph structure and labels.

**Understanding 2: Updating $\Theta$ is a pseudo-labeling approach.** Pseudo-labeling (Lee et al., 2013; Arazo et al., 2020) is a popular method in semi-supervised learning that uses a small set of labeled data along with a large amount of unlabeled data to improve model performance. It usually generates pseudo labels for the unlabeled data and trains the deep models using both the true labels and pseudo labels with different weights. From this perspective, Multiview4GNN uses the pseudo labels $\mathbf{F}$ to train $\Theta$ such that it can take advantage of both labeled and unlabeled nodes.

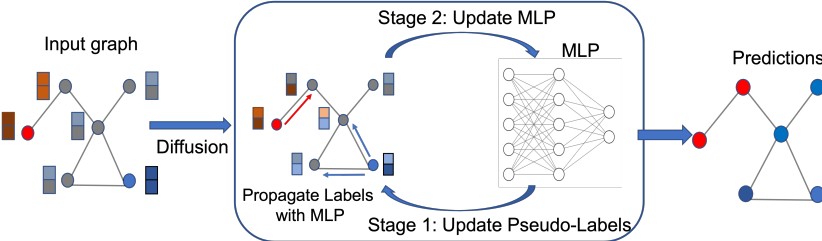

Figure 1: An overview of the proposed Multiview4GNN method, where grey color represents unlabeled nodes, and each node is associated with input features.

### 2.4 Implementation Details of Multiview4GNN

In this subsection, we detail the implementation of Multiview4GNN. As shown in Figure 1, we first preprocess the node feature through a diffusion, then alternatively update pseudo label $\mathbf{F}$ and MLP while taking into account the weight of pseudo labels and the class balancing problem, and finally get the prediction of unlabeled nodes. Next, we describe each step in detail.

**Preprocessing.** From Understanding 1, we use MLP to enhance the label propagation, so a good initialized MLP is needed. In real graphs, labeled data are usually scarce so that it is challenging to get a good initialization of MLP with a small number of labels. Therefore, we first diffuse the

original node features with its neighbors to get smoothing and enhanced features. The new features are obtained from $\mathbf{X}' = \text{LP}(\mathbf{X}, \alpha)$. Then, we train MLP only using the labeled data for a few epochs to get an initialization, similar to pseudo-labeling methods (Iscen et al., 2019; Lee et al., 2013).

**Update F.** We initial $\mathbf{F}^0 = \mathbf{Y}$. Then we update $\mathbf{F}$ for labeled nodes and unlabeled nodes by Eq. (7) and Eq. (8), respectively. Since $\mathbf{F}$ acts as pseudo labels when training the MLP, we normalize $\mathbf{F}$ to be the distribution of classes by using the softmax function with temperature after the update: $\mathbf{F}_{ij} = \frac{\exp(\mathbf{F}_{ij}/\tau)}{\sum_{k=1}^{C} \exp(\mathbf{F}_{ik}/\tau)}$, where $\tau$ is a hyperparameter to control the smoothness of pseudo labels.

**Pseudo-label certainty and class balancing.** Directly using all pseudo labels to train MLP is not appropriate due to the following reasons. First, not all pseudo labels have the same certainty. Second, pseudo-labels may not be balanced over classes, which will impede learning. To address the first issue, we assign a confidence weight to each pseudo-label. According to information theory, entropy can be used to quantify a distribution's uncertainty, so we define the weight for unlabeled nodes as $w_i = 1 - \frac{H(\mathbf{F}_i)}{\log(C)}$, where $w_i \in [0, 1]$ and $H(\mathbf{F}_i) = -\sum_{j=1}^{C} \mathbf{F}_{ij} \log \mathbf{F}_{ij}$ is the entropy of the pseudo label $\mathbf{F}_i$. To deal with the class imbalance problem, we adopt a simple method that chooses the same number of unlabeled nodes for each class with the highest weight to train MLP.

**Update MLP.** We train MLP using both labeled nodes set $L$ and high confidence unlabeled nodes set $U_t$. The loss function can be written as follows:

$$\mathcal{L}_{\text{MLP}}(\mathbf{X}', \mathbf{F}; \Theta) = \sum_{i \in L} \ell(\text{MLP}(\mathbf{X}'_i; \Theta), \mathbf{F}_i) + \sum_{j \in U_t} w_j \cdot \ell(\text{MLP}(\mathbf{X}'_j; \Theta), \mathbf{F}_j) \tag{10}$$

where $\ell(\text{MLP}(\mathbf{X}'_i; \Theta), \mathbf{F}_i) = \|\text{MLP}(\mathbf{X}'_i; \Theta) - \mathbf{F}_i\|_2^2$ is a MSE loss and $\Theta$ is the parameters of MLP.

**Prediction.** The inference of our method is based on the pseudo labels $\mathbf{F}$, and the predicted class for the unlabeled node $i$ can be obtained as $c_i = \arg\max_j \mathbf{F}_{ij}$. The overall algorithm, implementation details and code of MULTIVIEW4GNN are shown in Appendix B.

## 2.5 COMPLEXITY ANALYSIS

We provide time and memory complexity analyses for MULTIVIEW4GNN and the following representative GNNs: GCN (Kipf & Welling, 2016), SGC (Wu et al., 2019a), and APPNP (Klicpera et al., 2018). Suppose that $p$ is the number of propagation layers, $n$ is the number of nodes, $m$ is the number of edges, and $c$ is the number of classes. For simplicity, we assume that the hidden feature dimension is a fixed $d$ for all transformation layers, and we have $c \ll d$ in most cases; all feature transformations are updated $t$ epochs; Besides, the adjacent matrix $\mathbf{A}$ is a sparse matrix, and both forward and backward propagation have the same cost. Following (Li et al., 2021), we only analyze the inherent differences across models by assuming that all models have the same transformation layers, allowing us to disregard the time required for feature transformation and the memory footprint of network parameters. The time and memory complexities are summarized in Table 1.

**Time complexity.** We first analyze the time complexity of feature aggregation. The feature aggregation can be implemented as a sparse-dense matrix multiplication with cost $O(md)$ if the feature has $d$ dimensions. Therefore, the time complexity of training a $p$-layer GCN for $t$ epochs is $O(2tpmd)$ with the gradient backpropagation. For SGC, we only need $p$ steps of feature propagation, so

Table 1: Comparison of time and memory complexities.

| Method | Time | Memory |
|---|---|---|
| GCN | $O(2tpmd)$ | $O(nd + pnd)$ |
| SGC | $O(pmd)$ | $O(nd)$ |
| APPNP | $O(2tpmc)$ | $O(nd + pnc)$ |
| MULTIVIEW4GNN | $O(kpmc)$ | $O(nd + nc)$ |

the time complexity is $O(pmd)$. For APPNP, the gradient also needs to backpropagate through $p$ layers, but the feature dimension is $c$, resulting in the time complexity of $O(2tpmc)$. Regarding MULTIVIEW4GNN, as the model are optimized in an alternating way, there is no need to do both feature transformation and aggregation in each epoch. Rather, we can propagate the pseudo labels only for $k$ times during the whole training process. As a result, the time complexity of MULTIVIEW4GNN is $O(kpmc)$. In practice, choosing k from [2, 5] can achieve very promising performance, while $t$ needs to be 500 or 1,000 for other models to converge.

**Memory complexity.** It requires $O(nd)$ memory for storing node features. For the end-to-end training models, we need to store the intermediate state at each layer for gradient calculation. Specifically, for GCN, we need to store the hidden state for $p$ layers, so the memory complexity is $O((p+1)nd)$.

SGC only needs to store the propagated feature $O(nd)$ as we omit the memory of network parameters. Similarly, APPNP has the memory complexity of $O(nd + pnc)$. For MULTIVIEW4GNN, it does not need to store the gradients at each propagation layer. Instead, MULTIVIEW4GNN needs to hold the pseudo label $\mathbf{F}$. So the memory complexity of MULTIVIEW4GNN is $O(nd + nc)$.

If we omit the difference in the dimension of the propagation features (d = c), the time and memory of GCN and APPNP are the same, as they require feature propagation in each epoch. We call the methods that need propagation every epoch as **Persistent propagation methods.** Similarly, the methods that only need to propagate once, such as SGC, SIGN (Rossi et al., 2020), and C&S (Huang et al., 2020) have the same time and memory complexity, namely **One-time propagation methods**. MULTIVIEW4GNN is a **Lazy propagation method** since the features are propagated $k$ times during training with $k$ being a small number. Thus, MULTIVIEW4GNN can be seen as a balance between these two groups of methods.

## 3 EXPERIMENT

In this section, we verify the effectiveness of the proposed method, MULTIVIEW4GNN, through the semi-supervised node classification tasks. In particular, we try to answer the following questions:
- **RQ1:** How does MULTIVIEW4GNN perform when compared to other models?
- **RQ2:** Is MULTIVIEW4GNN more efficient than state-of-the-art GNNs?
- **RQ3:** How do different components affect MULTIVIEW4GNN?

### 3.1 EXPERIMENTAL SETTINGS

**Datasets.** For the transductive semi-supervised node classification task, we choose nine common used datasets including three citation datasets, i.e., Cora, Citeseer and Pubmed (Sen et al., 2008), two coauthors datasets, i.e., CS and Physics, two Amazon datasets, i.e., Computers and Photo (Shchur et al., 2018), and two OGB datasets, i.e., ogbn-arxiv and ogbn-products (Hu et al., 2020). For the inductive node classification task, we use Reddit and Flikcr datasets (Zeng et al., 2019). More details about these datasets are shown in Appendix C.

Following (Liu et al., 2021), we use 10 random data splits for the three citation datasets, and we run the experiments 3 times for each split. We report the average performance and standard deviation. Besides, we also test multiple labeling rates, i.e., 5, 10, 20, 60, 30% and 60% labeled nodes per class, to get a comprehensive comparison. For other datasets, we use the fixed split and run 10 times.

**Baselines.** We compare the proposed MULTIVIEW4GNN with three groups of methods: (i) Persistent propagation methods, i.e., GCN (Kipf & Welling, 2016), GAT (Veličković et al., 2017) and APPNP (Klicpera et al., 2018); (ii) One-time propagation methods, i.e., SGC (Wu et al., 2019a), SIGN Rossi et al. (2020), and C&S (Huang et al., 2020); and (iii) Non-GNN methods including MLP and Label Propagation(Zhou et al., 2003). We report test accuracy results of all models selected by the highest validation accuracy. Parameter settings for all methods are illustrated in Appendix E.

### 3.2 PERFORMANCE COMPARISON ON BENCHMARK DATASETS

**Transductive Node Classification.** The transductive node classification results are partially shown in Table 2. We leave results on more datasets and methods in Appendix D due to the space limitation. From these results, we can make the following observations:
- MULTIVIEW4GNN consistently outperforms other models at low label rates on all datasets. For example, in Cora and CiteSeer with label rate 5, our method can gain 1.2 % and 5.6 % relative improvement compared to the best baselines. This is because the pseudo labels generated by our framework are helpful for training models when there are few labels available. When the label rate is high, our method is also comparable to the best results. In addition, MULTIVIEW4GNN is alternately optimized but not end-to-end trained, which suggests that end-to-end training could not be necessary for node semi-supervised classification.
- MULTIVIEW4GNN performs the best on OGB datasets. For example, in ogbn-products, it obtains 7.86% and 2.63% relative improvement compared to APPNP and SIGN, respectively.
- Compared with the One-time propagation methods , Persistent propagation methods usually perform better when the labeling rate is low. In addition, the label propagation outperforms MLP in most cases, indicating the rationality of our proposed feature-enhanced label propagation.

Table 2: Transductive node classification accuracy (%) on benchmark datasets.

| Method | | Persistent propagation methods | | | One-time propagation methods | | | Ours |
|---|---|---|---|---|---|---|---|---|
| Dataset | Label | GCN | GAT | APPNP | SGC | SIGN | C&S | MULTIVIEW4GNN |
| Cora | 5 | 70.68 ± 2.17 | 72.97 ± 2.23 | 75.86 ± 2.34 | 70.06 ± 1.95 | 69.81 ± 3.13 | 56.52 ± 5.53 | **76.78 ± 2.56** |
| | 10 | 76.50 ± 1.42 | 78.03 ± 1.17 | 80.29 ± 1.00 | 76.28 ± 1.22 | 76.25 ± 1.26 | 71.04 ± 3.30 | **80.66 ± 1.92** |
| | 20 | 79.41 ± 1.30 | 81.39 ± 1.41 | 82.34 ± 0.67 | 80.30 ± 1.72 | 79.71 ± 1.11 | 77.96 ± 2.13 | **82.66 ± 0.98** |
| | 60 | 84.30 ± 1.44 | 85.11 ± 1.10 | 85.49 ± 1.25 | 84.17 ± 1.39 | 84.16 ± 1.18 | 82.21 ± 1.45 | **85.60 ± 1.12** |
| | 30% | 86.87 ± 1.35 | 87.24 ± 1.19 | **87.77 ± 1.13** | 86.97 ± 0.90 | 87.17 ± 1.28 | 87.60 ± 1.12 | 87.70 ± 1.19 |
| | 60% | 88.60 ± 1.19 | 88.68 ± 1.13 | 88.49 ± 1.28 | 88.60 ± 1.38 | 88.21 ± 1.11 | 88.68 ± 1.39 | **88.96 ± 1.10** |
| CiteSeer | 5 | 61.27 ± 3.85 | 62.60 ± 3.34 | 63.92 ± 3.39 | 60.21 ± 3.48 | 57.44 ± 3.71 | 50.39 ± 4.70 | **67.48 ± 2.90** |
| | 10 | 66.28 ± 2.14 | 66.81 ± 2.10 | 67.57 ± 2.05 | 65.23 ± 2.36 | 63.87 ± 3.09 | 58.96 ± 2.75 | **69.39 ± 2.59** |
| | 20 | 69.60 ± 1.67 | 69.66 ± 1.47 | 70.85 ± 1.45 | 68.82 ± 2.11 | 68.60 ± 1.94 | 65.85 ± 2.74 | **71.26 ± 1.69** |
| | 60 | 72.52 ± 1.74 | 73.10 ± 1.20 | **73.50 ± 1.54** | 71.43 ± 1.26 | 72.63 ± 1.39 | 71.21 ± 1.79 | 72.84 ± 1.65 |
| | 30% | 75.20 ± 0.85 | 75.01 ± 0.99 | **75.71 ± 0.71** | 75.09 ± 1.01 | 74.44 ± 0.83 | 74.65 ± 0.95 | 75.09 ± 0.79 |
| | 60% | 76.88 ± 1.78 | 76.70 ± 1.81 | **77.42 ± 1.47** | 76.66 ± 1.59 | 76.41 ± 1.96 | 76.34 ± 1.37 | 77.00 ± 1.67 |
| Pubmed | 5 | 69.76 ± 6.46 | 70.42 ± 5.36 | 72.68 ± 5.68 | 68.55 ± 6.88 | 66.52 ± 6.15 | 65.3 ± 6.02 | **73.51 ± 4.80** |
| | 10 | 72.79 ± 3.58 | 73.35 ± 3.83 | 75.53 ± 3.85 | 72.80 ± 3.55 | 71.32 ± 3.70 | 72.51 ± 3.75 | **75.55 ± 5.09** |
| | 20 | 77.43 ± 1.93 | 77.43 ± 2.66 | 78.93 ± 2.11 | 76.48 ± 2.84 | 76.39 ± 2.65 | 75.34 ± 2.49 | **79.16 ± 2.26** |
| | 60 | 82.00 ± 1.62 | 81.40 ± 1.40 | **82.55 ± 1.47** | 80.34 ± 1.61 | 81.75 ± 1.55 | 80.63 ± 1.49 | 82.53 ± 1.76 |
| | 30% | 88.07 ± 0.29 | 86.51 ± 0.41 | 87.56 ± 0.39 | 86.23 ± 0.43 | **89.09 ± 0.33** | 88.44 ± 0.40 | 88.24 ± 0.36 |
| | 60% | 88.48 ± 0.46 | 86.52 ± 0.56 | 87.56 ± 0.52 | 86.63 ± 0.38 | **89.55 ± 0.56** | 88.53 ± 0.56 | 88.83 ± 0.55 |
| ogbn-arxiv | 54% | 71.91 ± 0.15 | 71.92 ± 0.17 | 71.61 ± 0.30 | 68.74 ± 0.12 | 71.95 ± 0.11 | 71.03 ± 0.15 | **72.76 ± 0.17** |
| ogbn-products | 8% | 75.70 ± 0.19 | OOM | 76.62 ± 0.13 | 74.29 ± 0.12 | 80.52 ± 0.16 | 77.11 ± 0.06 | **82.64 ± 0.21** |

- The standard deviation of all models is not small across different data splits, especially when the label rate is very low. It demonstrates that splits can significantly affect a model's performance. A similar finding is also observed in the PyTorch-Geometric paper (Fey & Lenssen, 2019).

**Inductive Node Classification.** For inductive node classification, only training nodes can be observed in the graph during training, and all nodes can be used during the inference (Zeng et al., 2019). For MULTIVIEW4GNN, we first train an MLP with the training nodes' features and then do inference for the unlabeled node using the feature-enhanced label propagation in Eq (7) and Eq (8). As shown in Table 3, the MULTIVIEW4GNN outperforms other baselines on the inductive node classification task. The only difference between MULTIVIEW4GNN and MLP is the feature-enhanced label propagation, and the performance improvement can demonstrate its superiority.

Table 3: Inductive node classification accuracy (%).

| Method | MLP | GCN | APPNP | SGC | C&S | MULTIVIEW4GNN |
|---|---|---|---|---|---|---|
| Reddit | 62.84 | 93.30 | 94.11 | 93.85 | 95.30 | 95.74 |
| Flickr | 37.87 | 49.20 | 49.40 | 50.58 | 51.46 | 52.29 |

## 3.3 EFFICIENCY COMPARISON

In this subsection, we compare the efficiency of our MULTIVIEW4GNN with other baselines, based on two large datasets, i.e., ogbn-arxiv and ogbn-products. To make a fair comparison, we choose the identical feature transformation layers for each method as there are no learnable parameters in feature propagation layers. Besides, we train model parameters with the same iterations in each method, i.e., 500 epochs for ogbn-arxiv and 1,000 epochs for ogbn-products. All the experiments are conducted on the same machine with a NVIDIA RTX A6000 GPU (48 GB memory).

For MULTIVIEW4GNN, we can update $\mathbf{F}$ with different frequency in training, i.e., 1, 2, 3, 4, 5, and "Full". "Full" means we update both $\mathbf{F}$ and MLP in each epoch. For MULTIVIEW4GNN-$k$, we only update the $\mathbf{F}$ for $k$ times during the training procedure. The overall results are shown in Table 4.

Table 4: Efficiency comparison of different methods.

| Dataset | ogbn-arxiv | | | ogbn-products | | |
|---|---|---|---|---|---|---|
| Method | ACC(%) | Time (s) | Memory (GB) | ACC (%) | Time (s) | Memory (GB) |
| MLP | 55.68 | 12.01 | 2.68 | 61.17 | 214 | 21.18 |
| SGC | 66.92 | 12.06 | 2.71 | 74.29 | 215 | 21.85 |
| SIGN | 71.95 | 24.89 | 4.67 | 80.52 | 492 | 43.17 |
| GCN | 71.91 | 24.71 | 3.33 | 75.70 | 1,284 | 38.36 |
| APPNP | 71.61 | 33.70 | 3.20 | 76.62 | 1,913 | 29.15 |
| MULTIVIEW4GNN-Full | 72.76 | 22.28 | 2.81 | 81.83 | 901 | 24.49 |
| MULTIVIEW4GNN-1 | 70.09 | 12.86 | 2.81 | 80.03 | 218 | 24.49 |
| MULTIVIEW4GNN-2 | 72.32 | 12.89 | 2.81 | 80.34 | 219 | 24.49 |
| MULTIVIEW4GNN-3 | 72.60 | 12.92 | 2.81 | 81.00 | 220 | 24.49 |
| MULTIVIEW4GNN-4 | 72.71 | 12.95 | 2.81 | 81.89 | 221 | 24.49 |
| MULTIVIEW4GNN-5 | 72.70 | 12.98 | 2.81 | 82.64 | 222 | 24.49 |

**Training Time.** For the Persistent propagation methods including GCN, APPNP and MULTI-VIEW4GNN-Full, the training time is longer than other methods that do not need to propagate every epoch. Both APPNP and MULTIVIEW4GNN-Full need to propagate ten layers every epoch and GCN needs to propagate three layers. However, the training time of MULTIVIEW4GNN-Full is nearly half of APPNP and still less than GCN, which matches our time complexity analysis in Section 2.5, as there is no gradient backpropagate through propagation layers. Compared with the One-time propagation methods like SGC, MULTIVIEW4GNN with only a few update steps, such as MULTIVIEW4GNN-5, can achieve better accuracy with a minor increase in training time. For example, the whole training time of MULTIVIEW4GNN-5 is only 0.92s and 7s longer than SGC, but it has 8.63% and 11.24 % relative performance improvements in ogbn-arxiv and ogbn-products datasets, respectively. Meanwhile, we can observe that MULTIVIEW4GNN-5 has very similar performance with MULTIVIEW4GNN-Full, which suggests that there is no need to do propagation and train the model simultaneously for each epoch. This also suggests that the end-to-end training with propagation might not be necessary.

**Memory Cost.** Compared with the Persistent propagation methods, MULTIVIEW4GNN requires less memory with no requirement to store the hidden states in the propagation layers. Thus, MULTIVIEW4GNN can keep a constant memory even with more propagation layers. Compared with MLP and SGC, MULTIVIEW4GNN only slightly increases memory as it needs to store the pseudo label matrix as analyzed in Section 2.5. In addition, MULTIVIEW4GNN is suitable for large-scale datasets that cannot fit in the GPU memory. First, it is easy to propagate the pseudo labels using CPUs as the label dimension is often lower than that of features. Then, MULTIVIEW4GNN is amenable to sampling training with mini-batch, which would significantly reduce the memory cost as discussed in Section 2.2. The only additional cost beyond One-time propagation methods is that we need to transfer the result of MLP from GPU to CPU to do feature enhanced label propagation. Due to the small dimension of labels and the limited number of propagations during training, the cost is negligible.

## 3.4 ABLATION STUDY

In this subsection, we conduct ablation studies to gain a better understanding of how each component of our method works which correspondingly answers the third question.

**Feature Diffusion.** It is expected that feature diffusion can improve the accuracy of the MLP in the pretraining procedure when the label rate is low and thus improve the quality of pseudo labels $\mathbf{F}$ during its following update steps. To validate this, we remove the feature diffusion step and move the feature diffusion step and

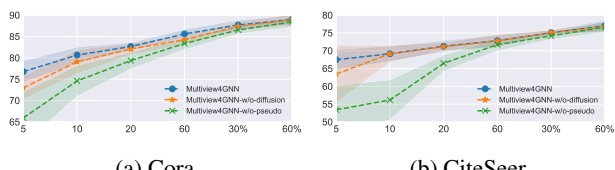

(a) Cora        (b) CiteSeer

Figure 2: Performance of MULTIVIEW4GNN variants.

also use the pseudo labels to train our method, which is called MULTIVIEW4GNN-w/o-diffusion. Experiments are conducted on both Cora and CiteSeer datasets. From Figure 2, we can see that at low label rates, MULTIVIEW4GNN is better than MULTIVIEW4GNN-w/o-diffusion which means that feature diffusion can boost the model's performance on low label rate setting. As the labeling rate increases, the performance gap becomes small, especially in the CiteSeer dataset. This shows that feature diffusion is not the key component in our method when the label rate is not very low.

**Pseudo Labels.** One of the most important advantages of MULTIVIEW4GNN is that we leverage pseudo labels to better train MLP. To study the contribution of pseudo labels in MULTIVIEW4GNN, we test the model variant MULTIVIEW4GNN-w/o-pseudo which only uses labeled data on Cora and CiteSeer datasets. Compared with MULTIVIEW4GNN, Figure 2 shows that pseudo labels have a large impact on model performance on both datasets, especially when the label rate is low.

Moreover, we choose the top $K$ confidence pseudo labels per class after the first update of $\mathbf{F}$ to verify their accuracy. We adopt the same way to evaluate Label Propagation on Cora dataset with the label rate 20. As shown in Figure 3, after the first update of $\mathbf{F}$, the accuracy of the top 180 nodes from each class can be $90\%$. So it is reasonable to use these pseudo labels to train MLP. Besides, the accuracy of our method at each K is much better than Label propagation, which suggests the effectiveness of the feature-enhanced label propagation update for $\mathbf{F}$.

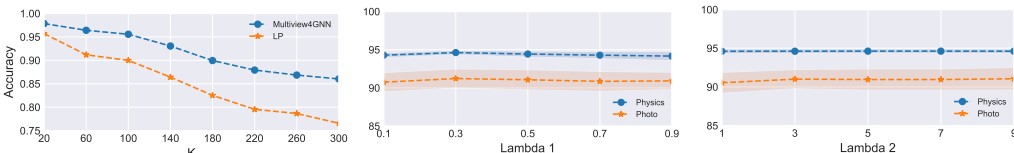

Figure 3: Top K Accuracy.                    Figure 4: Parameter Sensitivity.

**Hyperparameters Sensitivity.** We test the parameter sensitivity of $\lambda_1$ and $\lambda_2$ in Eq (2) on Physics and Photo datasets by fixing one with the best parameters and tuning the other. From Figure 4, MULTIVIEW4GNN is not very sensitive to these two hyperparameters at the chosen regions.

## 4 RELATED WORK

Graph Neural Network (GNN) is an effective architecture to represent the graph-structure data. Two essential operations in the GNN are feature propagation and feature transformation. Considering how may times feature propagation in the training procedure, we categorize GNNs into Persistent propagation GNNs and One-time propagation GNNs. Persistent propagation GNNs (e.g., GCN (Kipf & Welling, 2016), GraphSAGE (Hamilton et al., 2017), GAT (Velickovic et al., 2017)) require feature propagation on each training step. GCN (Kipf & Welling, 2016) is the most commonly used method which performs feature transformation then feature aggregation in each layer. More recently, decoupled GNNs are proposed to alleviate the over-smoothness problem (Li et al., 2018; Oono & Suzuki, 2019). APPNP (Klicpera et al., 2018) is the first work to first apply multiple feature transformations then multiple aggregations. Similar architectures are also utilized in (Liu et al., 2021; 2020; Zhou et al., 2021). However, these decoupled GNNs still require feature aggregation in each training step. One-time propagation GNNs are more efficient than the above Persistent propagation methods for they only propagate once despite the number of training steps. SGC (Wu et al., 2019a) is the first one to do feature aggregation and then transformation. SIGN (Rossi et al., 2020) adopts a similar strategy with a different aggregation scheme. Recently, more efficient neural networks are proposed which utilize graph structure for post-process. Huang et al. (2020) train a base predictor on labeled data and then apply a correct and a smooth step to post-process. Dong et al. (2021) further understand the relationship between decoupled GNNs and label propagation and utilizes soft pseudo labels for training. PPRGo (Bojchevski et al., 2020) precomputes the PageRank matrix but it lose much edge information due to aggressive sparsification of the PageRank matrix, which often leads to degraded performance and non-trivial tradeoff between efficiency and accuracy.

**Multi-view learning on Graph.** Our proposed multi-view learning framework for graph representation learning differs from existing works in literature. Specifically, multi-view graph cluster (Wang et al., 2019; Pan & Kang, 2021) consider a totally different setting where multi-view attributes and multiple structural graphs exist. Multi-view graph contrastive learning algorithms (Hassani & Khasahmadi, 2020; Wang et al., 2021) use data augmentation to generate different graph views, then encourage the similarity between different views generated from the same graph while reduce the similarity in other view pairs. In contrast, our multi-view learning framework considers node features, graph structure, and node labels as three views of the nodes.

**Unified understanding on GNN.** Recent works (Zhu et al., 2021; Ma et al., 2021; Yang et al., 2021) aim to provide a unified optimization framework for understanding the message passing mechanism of different GNNs and designing new graph filter layers. However, they only focus on the forward process without taking the backward learning process into consideration, and they are still following the existing GNN architecture with end-to-end training. In this work, we do not aim to understand the message passing and design new layers based on existing architectures. Instead, MULTIVIEW4GNN is a new graph deep learning framework as multi-view learning. It provides a new perspective for graph representation learning with better flexibility, explanability, and efficiency.

## 5 CONCLUSION

In this work, we provide a new perspective to view the three types of information available for node classification (i.e., graph structure, node feature, and node label) as three views of nodes. This understanding inspires us to design GNN models as multi-view learning. The proposed MULTIVIEW4GNN framework can naturally be trained with the alternating optimization algorithm. Experimental results validate that MULTIVIEW4GNN is both computational and memory efficient with promising performance on the node classification task especially when the label rate is low.

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

## A   MULTIVIEW4GNN WITH CROSS ENTROPY LOSS

In section 2.2, we instance MULTIVIEW4GNN with a Mean Square Error Loss. In this section, we show it can be replaced by a Cross Entropy Loss. By replacing the first part of Eq (2) to be a cross entropy between MLP and $\mathbf{F}$, the formulation becomes:

$$\mathcal{L} = \lambda_1 CE\left(\text{MLP}(\mathbf{X}), \mathbf{F}\right) + \text{tr}(\mathbf{F}^\top \tilde{\mathbf{L}} \mathbf{F}) + \lambda_2 \|\mathbf{F}_L - \mathbf{Y}_L\|_F^2 \tag{11}$$

where $CE(\cdot, \cdot)$ is the cross entropy function. Adopting the same gradient decent method, the update rule for $\mathbf{F}$ becomes:

$$\mathbf{F}_L^{k+1} = \mathbf{F}_L^k - \eta_L \left(-\lambda_1 \log \text{MLP}(\mathbf{X}_L) + 2(\tilde{\mathbf{L}}\mathbf{F}^k)_L + 2\lambda_2(\mathbf{F}_L^k - \mathbf{Y}_L)\right)$$

$$= (1 - 2\eta_L(1 + \lambda_2))\mathbf{F}_L^k + \eta_L \lambda_1 \log \text{MLP}(X)_L + 2\eta_L \tilde{\mathbf{A}}\mathbf{F}_L^k + 2\eta_L \lambda_2 \mathbf{Y}_L, \tag{12}$$

$$\mathbf{F}_U^{k+1} = \mathbf{F}_U^k - \eta_U \left(-\lambda_1 \log \text{MLP}(\mathbf{X}_U) + 2(\tilde{\mathbf{L}}\mathbf{F}^k)_U\right)$$

$$= (1 - 2\eta_U)\mathbf{F}_U^k + \eta_U \lambda_1 \log \text{MLP}(\mathbf{X}_U) + 2\eta_U(\tilde{\mathbf{A}}\mathbf{F}^k)_U \tag{13}$$

where the MLP is the output after the Softmax function and the step size can be set as $\eta_L = \eta_U = \frac{1}{2(1+\lambda_2)}$. Therefore, the update rule of $\mathbf{F}$ is:

$$\mathbf{F}_L^{k+1} = \frac{\lambda_1}{2(1+\lambda_2)} \log \text{MLP}(\mathbf{X}_L) + \frac{1}{1+\lambda_2}(\tilde{\mathbf{A}}\mathbf{F}^k)_L + \frac{\lambda_2}{1+\lambda_2}\mathbf{Y}_L, \tag{14}$$

$$\mathbf{F}_U^{k+1} = \frac{\lambda_1}{2(1+\lambda 2)} \log \text{MLP}(\mathbf{X}_U) + \frac{1}{1+\lambda_2}(\tilde{\mathbf{A}}\mathbf{F}^k)_U + \frac{\lambda_2}{1+\lambda_2}\mathbf{F}_U^k \tag{15}$$

Then we can consider the hidden variable $\mathbf{F}$ as pseudo label and update the parameters of MLP based on the cross entropy loss $CE(\text{MLP}, \mathbf{F})$. In practice, there are some situations that the cross entropy have a better performance than the original mean square error loss. Using two different losses would give similar results in most cases, and we report the best one.

## B   ALGORITHM OF MULTIVIEW4GNN

In this section, we provide the algorithm 1 and code of MULTIVIEW4GNN.

In line 1, we first initialize the pseudo label $\mathbf{F}$ as label matrix $\mathbf{Y}$. In line 2, we preprocess data by feature diffusion. In line 3, we pretrain the MLP on the labeled data for a few epochs. From lines 4 to 9, we update $\mathbf{F}$ and MLP alternatively and iteratively.

---

**Algorithm 1:** MULTIVIEW4GNN

**Input**  : Adjacent matrix $\mathbf{A}$; Feature matrix $\mathbf{X}$; Labels $\mathbf{Y}$; Hyper-parameters $\lambda_1, \lambda_2, \alpha$, pseudo node number $m$; Labeled nodes $L$; layer $p$, pretraining step S; update times $t$

**Output** : Pseudo Label $\mathbf{F}$, MLP parameter $\Theta$

1   Initialize $\mathbf{F} \leftarrow \mathbf{Y}$ ;
2   $\mathbf{X}' \leftarrow LP(\mathbf{X}, \alpha)$ ;
3   Pretraining S steps MLP with $\mathcal{L} = \sum_{i \in L} \ell(\text{MLP}(x_i'; \Theta), \mathbf{Y}_i)$;
4   **while** *Stopping condition is not met* **do**
5   $\quad$ Update $\mathbf{F}$ based on Eq. (7) and Eq. (8) for $p$ times ;
6   $\quad$ Normalize $\mathbf{F}$ based on $\mathbf{F}_{ij} = \frac{\exp(\mathbf{F}_{ij}/\tau)}{\sum_{k=1}^C \exp(\mathbf{F}_{ik}/\tau)}$ ;
7   $\quad$ Select $m$ top unlabeled nodes $U_t$ per class by $w_i = 1 - \frac{H(\mathbf{F}_i)}{log(C)}$;
8   $\quad$ Update $\Theta$ for $t$ times by minimizing $\mathcal{L}_{MLP}(X', F; \Theta) = \sum_{i \in L \cup U_t} \ell(\text{MLP}(x_i'; \Theta), \mathbf{F}_i)$ ;
9   **end**
10  **return** $\mathbf{F}, \Theta$ ;

---

The code implemented by Pytorch is avaliable at https://anonymous.4open.science/r/multiview4gnn-F418/.

## C    DATASET STATISTICS

In the experiments, the data statistics used in Section 3 are summarized in Table 5. For Cora, Cite-Seer and PubMed dataset, we adopt different label rates, i.e., 5, 10, 20, 60, 30% and 60% labeled nodes per class, to get a more comprehensive comparison. For label rates 5, 10, 20, and 60, we use 500 nodes for validation and 1000 nodes for test. For label rates 30% and 60%, we use half of the rest nodes for validation and the remaining half for test. For Ogbn-arxiv dataset, we use the original fixed data split.

Table 5: Dataset Statistics.

| Dataset | Nodes | Edges | Features | Classes |
|---|---|---|---|---|
| Cora | 2,708 | 5,278 | 1,433 | 7 |
| CiteSeer | 3,327 | 4,552 | 3,703 | 6 |
| PubMed | 19,717 | 44,324 | 500 | 3 |
| Coauthor CS | 18,333 | 81,894 | 6,805 | 15 |
| Coauthor Physics | 34,493 | 247,962 | 8,415 | 5 |
| Amazon Computer | 13,381 | 245,778 | 767 | 10 |
| Amazon Photo | 7,487 | 119,043 | 745 | 8 |
| Flickr | 89,250 | 899,756 | 500 | 7 |
| Reddit | 232,965 | 11,606,919 | 602 | 41 |
| Ogbn-Arxiv | 169,343 | 1,166,243 | 128 | 40 |
| Ogbn-Products | 2,449,029 | 61,859,140 | 100 | 47 |

## D    TRANSDUCTIVE NODE CLASSIFICATION RESULTS

Table 6: The overall results of the transductive node classification task.

| Method | | Non-GNN | | Persistent propagation methods | | | One-time propagation methods | | | Ours |
|---|---|---|---|---|---|---|---|---|---|---|
| Dataset | Label Rate | LP | MLP | GCN | GAT | APPNP | SGC | SIGN | C&S | Multiview4GNN |
| Cora | 5 | 57.60 ± 5.71 | 42.34 ± 3.31 | 70.68 ± 2.17 | 72.97 ± 2.23 | 75.86 ± 2.34 | 70.06 ± 1.95 | 69.81 ± 3.13 | 56.52 ± 5.53 | **76.78 ± 2.56** |
| | 10 | 63.76 ± 3.60 | 51.34 ± 3.37 | 76.50 ± 1.42 | 78.03 ± 1.17 | 80.29 ± 1.00 | 76.28 ± 1.22 | 76.25 ± 1.26 | 71.04 ± 3.30 | **80.66 ± 1.92** |
| | 20 | 67.87 ± 1.43 | 59.23 ± 2.52 | 79.41 ± 1.30 | 81.39 ± 1.41 | 82.34 ± 0.67 | 80.30 ± 1.72 | 79.71 ± 1.11 | 77.96 ± 2.13 | **82.66 ± 0.98** |
| | 60 | 73.92 ± 1.25 | 68.35 ± 2.08 | 84.30 ± 1.44 | 85.11 ± 1.10 | 85.49 ± 1.25 | 84.17 ± 1.39 | 84.16 ± 1.18 | 82.21 ± 1.45 | **85.60 ± 1.12** |
| | 30% | 82.26 ± 1.89 | 73.26 ± 1.38 | 86.87 ± 1.35 | 87.24 ± 1.19 | **87.77 ± 1.13** | 86.97 ± 0.90 | 87.17 ± 1.28 | 87.60 ± 1.12 | 87.70 ± 1.19 |
| | 60% | 86.05 ± 1.35 | 76.49 ± 1.13 | 88.60 ± 1.19 | 88.68 ± 1.13 | 88.49 ± 1.28 | 88.60 ± 1.38 | 88.21 ± 1.11 | 88.68 ± 1.39 | **88.96 ± 1.10** |
| CiteSeer | 5 | 39.06 ± 3.53 | 41.05 ± 2.84 | 61.27 ± 3.85 | 62.60 ± 3.34 | 63.92 ± 3.39 | 60.21 ± 3.48 | 57.44 ± 3.71 | 50.39 ± 4.70 | **67.48 ± 2.90** |
| | 10 | 42.29 ± 3.26 | 47.99 ± 2.71 | 66.28 ± 2.14 | 66.81 ± 2.10 | 67.57 ± 2.05 | 65.23 ± 2.36 | 63.87 ± 3.09 | 58.96 ± 2.75 | **69.39 ± 2.59** |
| | 20 | 46.15 ± 2.31 | 56.96 ± 1.80 | 69.60 ± 1.67 | 69.66 ± 1.47 | 70.85 ± 1.45 | 68.82 ± 2.11 | 68.60 ± 1.94 | 65.85 ± 2.74 | **71.26 ± 1.69** |
| | 60 | 52.76 ± 1.14 | 66.37 ± 1.56 | 72.52 ± 1.74 | 73.10 ± 1.20 | **73.50 ± 1.54** | 71.43 ± 1.26 | 72.63 ± 1.39 | 71.21 ± 1.79 | 72.84 ± 1.65 |
| | 30% | 62.75 ± 1.30 | 70.37 ± 1.00 | 75.20 ± 0.85 | 75.01 ± 0.99 | **75.71 ± 0.71** | 75.09 ± 1.01 | 74.44 ± 0.83 | 74.65 ± 0.95 | 75.09 ± 0.79 |
| | 60% | 69.39 ± 2.01 | 73.15 ± 1.36 | 76.88 ± 1.78 | 76.70 ± 1.81 | **77.42 ± 1.47** | 76.66 ± 1.59 | 76.41 ± 1.96 | 76.34 ± 1.37 | 77.00 ± 1.67 |
| Pubmed | 5 | 65.52 ± 6.42 | 58.48 ± 4.06 | 69.76 ± 6.46 | 70.42 ± 5.36 | 72.68 ± 5.68 | 68.55 ± 6.88 | 66.52 ± 6.15 | 65.3 ± 6.02 | **73.51 ± 4.80** |
| | 10 | 68.39 ± 4.88 | 65.36 ± 2.08 | 72.79 ± 3.58 | 73.35 ± 3.83 | 75.53 ± 3.85 | 72.80 ± 3.55 | 71.32 ± 3.70 | 72.51 ± 3.75 | **75.55 ± 5.09** |
| | 20 | 71.88 ± 1.72 | 69.07 ± 2.10 | 77.43 ± 1.93 | 77.43 ± 2.66 | 78.93 ± 2.11 | 76.48 ± 2.84 | 76.39 ± 2.65 | 75.34 ± 2.49 | **79.16 ± 2.26** |
| | 60 | 75.79 ± 1.54 | 76.20 ± 1.48 | 82.00 ± 1.62 | 81.40 ± 1.40 | **82.55 ± 1.47** | 80.34 ± 1.61 | 81.75 ± 1.55 | 80.63 ± 1.49 | 82.53 ± 1.76 |
| | 30% | 82.51 ± 0.34 | 85.92 ± 0.25 | 88.07 ± 0.29 | 86.51 ± 0.41 | 87.56 ± 0.39 | 86.23 ± 0.43 | 89.09 ± 0.33 | 88.44 ± 0.40 | 88.24 ± 0.36 |
| | 60% | 83.38 ± 0.64 | 86.14 ± 0.64 | 88.48 ± 0.46 | 86.52 ± 0.56 | 87.56 ± 0.52 | 86.63 ± 0.38 | 89.55 ± 0.56 | 88.53 ± 0.56 | **88.83 ± 0.55** |
| CS | 20 | 77.45 ± 1.80 | 88.12 ± 0.78 | 91.73 ± 0.49 | 90.96 ± 0.46 | 92.38 ± 0.38 | 90.32 ± 0.99 | 92.02 ± 0.41 | 92.41 ± 0.44 | **92.77 ± 0.50** |
| Physics | 20 | 86.70 ± 1.03 | 88.30 ± 1.59 | 93.29 ± 0.80 | 92.81 ± 1.03 | 93.49 ± 0.67 | 93.23 ± 0.59 | 93.03 ± 1.15 | 93.23 ± 0.55 | **94.63 ± 0.31** |
| Computers | 20 | 72.44 ± 2.87 | 60.66 ± 2.98 | 79.17 ± 1.92 | 78.38 ± 2.27 | 79.07 ± 2.34 | 73.00 ± 2.0 | 73.04 ±1.15 | 73.25± 2.09 | **79.12 ± 2.50** |
| Photo | 20 | 81.58 ± 4.69 | 75.33 ± 1.91 | 89.94 ± 1.22 | 89.24 ± 1.42 | 90.87 ± 1.14 | 83.50 ± 2.9 | 86.11 ± 0.66 | 84.87 ± 1.04 | **91.23 ± 1.26** |
| ogbn-arxiv | 54% | 68.14 ± 0.00 | 55.68 ± 0.11 | 71.91 ± 0.15 | 71.92 ± 0.17 | 71.61 ± 0.30 | 68.74 ± 0.12 | 71.95 ± 0.11 | 71.03 ± 0.15 | **72.76 ± 0.17** |
| ogbn-products | 8% | 74.08 ± 0.00 | 61.17 ± 0.20 | 75.70 ± 0.19 | OOM | 76.62 ± 0.13 | 73.15 ± 0.12 | 80.52±0.16 | 77.11 ± 0.06 | **82.64± 0.21** |

## E    PARAMETER SETTING

### E.1    TRANSDUCTIVE SETTINGS

For all deep models, we use 3 transformation layers with 256 hidden units for OGB datasets and 2 layers with 64 hidden units for other datasets. We use the same learning rate of 0.01. For all methods, hyperparameters are tuned based on the loss and validation accuracy from the following search space: 1) dropout rate: $\{0, 0.5, 0.8\}$; 2) weight decay: $\{0, 5e\text{-}4, 5e\text{-}5\}$; and 3) For the hyperparameters range between 0 and 1, we tune them by step size 0.1. The propagation step $K$ for APPNP and C&S is tuned from $\{5, 10\}$ and $\{10, 20, 50\}$, respectively. The $\lambda_1$ and $\lambda_2$ in MULTIVIEW4GNN are tuned from $\{0.1, 0.3, 0.5, 0.7, 1\}$ and $\{1, 3, 5, 7, 10\}$, respectively. Adam optimizer(Kingma & Ba, 2014) is used in all experiments.

### E.2 INDUCTIVE SETTINGS

We first filter the training graph that only contains labeled node for training, and the entire graph are used for inference. For all models, we use 3 transformation layers with 256 hidden units for Reddit dataset, and 2 layers with 64 hidden units for Flickr dataset. For all methods, hyperparameters are tuned based on the loss and validation accuracy from the following search space: (1) learning rate: {0.01, 0.05 }; (2) dropout: {0, 0.1}; (3) weight decay: {0}; For the hyperparameters range between 0 and 1, we tune them by step size 0.1. The propagation step K for APPNP and C& is tuned from {2, 3, 5, 10} and {10, 20, 50}, respectively. The $\lambda_1$ and $\lambda_2$ in MULTIVIEW4GNN are tuned with granularity of 0.1 in range [0, 1], and 1 in [1, 10], respectively.

