# OpenReview forum: "Graph Neural Networks as Multi-View Learning"
_ICLR.cc/2023/Conference — Submitted to ICLR 2023_

### Official Review · Reviewer_xFqB · 2022-10-24

**Confidence:** 4
**Correctness:** 3
**Technical Novelty And Significance:** 2
**Empirical Novelty And Significance:** 2
**Recommendation:** 5

**Clarity, Quality, Novelty And Reproducibility:**

This paper clearly presents the multi-view loss design and provides solid experiment results. The technical novelty of the proposed method is a bit limited.

**Strength And Weaknesses:**

Strength:

1. The proposed multi-view loss (with *lazy propagation*) is an effective method to achieve the trade-off between performance and cost.
2. The authors conduct extensive ablation experiments to verify each model component.

Weaknesses:

1. The three losses after decoupling are not novel individually. Each component seems to be standard and well-studied in graph semi-supervised learning.
2. The pseudo-label with certainty and class-balancing in graph semi-supervised learning are also discussed in [1]. And there are other works highly related to the core ideas in this paper, e.g., [2,3,4], should be discussed.

References:

[1]. Self-Enhanced GNN: Improving Graph Neural Networks Using Model Outputs. Han Yang, Xiao Yan, Xinyan Dai, Yongqiang Chen, James Cheng. IJCNN '2021

[2]. Rethinking and Scaling Up Graph Contrastive Learning: An Extremely Efficient Approach with Group Discrimination. Yizhen Zheng, Shirui Pan, Vincent Cs Lee, Yu Zheng, Philip S. Yu. NeurIPS ’22.

[3]. Combining Label Propagation and Simple Models Out-performs Graph Neural Networks. Qian Huang, Horace He, Abhay Singh, Ser-Nam Lim, Austin R. Benson. ICLR ’21.

[4]. On Graph Neural Networks versus Graph-Augmented MLPs. Lei Chen, Zhengdao Chen, Joan Bruna. ICLR ’21.


**Summary Of The Paper:**

This paper considers node features, graph structure, and node labels as three views, and designs three loss components correspondingly. Moreover, the latent $F$ and the model parameters $\Theta$ are proposed to be optimized alternatingly to achieve efficiency and scalability.


**Summary Of The Review:**

This paper decouples the supervised loss and optimizes it alternatingly to achieve efficiency and scalability. However, the technical contribution is a bit limited.

---

> ### Author Response · Authors · 2022-11-19
> **Response to Reviewer xFqB**
>
> Dear reviewer,
>
> Thanks for the valuable feedback. Here we provide detailed responses to address the concerns in your comments.
>
> Q1: The three losses after decoupling are not novel individually. Each component seems to be standard and well-studied in graph semi-supervised learning.
>
> Answer: The three losses is one instance of our unified framework, and there are numerous designs from this framework as shown in Section 2.1.
> The instance can be represented as  $L =  \lambda_1{||MLP(X)-F||_F^2} + tr(F^\top L F) + \lambda_2 {||F_L-Y_L||_F^2}$.  If we only consider the first two parts of $L$, i.e., $L_1 = min_F \lambda_1{||MLP(X)-F||_F^2} + {tr(F^\top L F)}$, it is the optimization target for many GNNs [1], such as GCN, GAT and APPNP. The forward process (feature propagation) of these GNNs is to optimize $L_1$, and these GNNs can be used in various tasks, such as node classification, link prediction, and graph classification. However, if we only consider the node classification task, the label information is only used in the MSE loss $L_M= min_F ||F_L-Y_L||_F^2$ (or Cross Entropy loss).  **As a result, the optimization for $L_1$ and optimization for $L_M$ are two separate targets.** The forward process (optimize $L_1$) is fixed once we choose the model architecture, for example, the teleportation parameter $\alpha$ and propagation layers of APPNP. After the fixed optimization for $L_1$, we  optimize $L_M$ using labels. **These two optimizations are independent and there might be some conflicts.** If we only consider the last two parts, i.e.,  $L_2 = min_F {tr(F^\top L F)} + \lambda_2 {||F_L-Y_L||_F^2}$, it is the label propagation. Label Propagation uses the label information in the forward process for classification and have only this one optimization target.
>
> Instead of first designing GNNs from $L_1$, and then applying them to specific tasks, in this paper, we change our perspective by considering the node feature, graph structure, and labels in the node classification task as three views. Then, we can find the proposed multi-view learning framework actually integrate the two optimization targets in GNNs for node classification into a single optimization objective. As a result, the two optimization targets can be concurrently optimized in our framework, instead of first optimizing the $L_1$ loss and then optimizing $L_M$ in GNNs.
>
> Another advantage of our framework is efficiency. Our method can achieve comparable performance but significantly better computation and memory efficiency comparing with Persistent propagation methods, such as GCN and APPNP. From Table 4 in our paper, the training time of Multiview4GNN is slightly more than MLP on both ogbn-arxiv and ogbn-products dataset. The training time of APPNP on the ogbn-products dataset is 9 times that of Multiview4GNN. Meanwhile, the memory cost is also less than GCN and APPNP. In addition, our method only need to train an MLP, which is simple to do batch-training. It can significantly reduces the memory cost.
>
> [1] Yao Ma, et al. A unified view on graph neural networks as graph signal denoising. CIKM'21
>
> Q2: The pseudo-label with certainty and class-balancing in graph semi-supervised learning are also discussed in [1]. And there are other works, e.g., [2,3,4], should be discussed.
>
> Thanks you for these related works. In the literature, the pseudo label selection and re-weighting are two widely used tricks in Pseudo Labeling approaches [5, 6]. The discussions of paper [1, 2, 3, 4] are shown below:
>
> * [1] also generates some pseudo labels for training node augmentation. However, [1] generated pseudo labels by the diversity of multiple GNN models, while our method use the feature enhanced label propagation to generate pseudo labels. In our model, we only need to train an MLP, which is more efficient than [1].
>
> * [2] proposes an efficient graph contrastive learning method using Group Discrimination, which simplify the discriminator in DGI. However, [2] is an end-to-end training model and there is propagation in each epoch.
>
> * [3] is a One-time propagation method. It first trains an MLP and then conducts postprocess such as correct and smooth. Different from [3], our model can propagate $k$ times during training and generate pseudo labels to train MLP better.
>
> * [4] first augments nodes features based on the graph topology, and then trains an MLP based on the new features. [4] is kind of similar to SGC, and it only generate the new features once. However, our Multiview4GNN can generate the new pseudo labels k times.
>
> [5] Lee, Dong-Hyun. "Pseudo-label: The simple and efficient semi-supervised learning method for deep neural networks." ICML'13.
>
> [6]Rizve, Mamshad Nayeem, et al. "In defense of pseudo-labeling: An uncertainty-aware pseudo-label selection framework for semi-supervised learning. ICLR'21.
>
> Overall, we hope that we have addressed the concerns, and please kindly let us know if there is any further concern, and we are happy to clarify.

---

### Official Review · Reviewer_ogUi · 2022-10-24

**Confidence:** 4
**Correctness:** 2
**Technical Novelty And Significance:** 2
**Empirical Novelty And Significance:** 2
**Recommendation:** 3

**Clarity, Quality, Novelty And Reproducibility:**

Clarity and Quality:
  - The paper is well presented with clear notations and references. The experiment setups are reasonable.

Novelty:
  - Not too much, because the whole training scheme is very similar to the Iscen et al. 2019 "Label Propagation for Deep Semi-supervised Learning" paper, including most of the important training tricks.

**Strength And Weaknesses:**

Strength:
  - The paper is well presented, with good support materials in the Appendix, including speudo code, dataset statistics, and parameter setting.
  - The paper conducts relevant experiments, including ablation study and efficiency comparison.

Weaknesses:
  - The whole training scheme is very similar to the Iscen et al. 2019 "Label Propagation for Deep Semi-supervised Learning" paper, including most of the important training tricks.
  - Experiment results only show good improvement on very low label setting (5-60 examples per class). For most of the time, the improvements are not statistically significant.
  - Hyper-param search space for the proposed method is much larger than baseline models. It includes $\lambda_1$ and $\lambda_2$, which together create 25 to 100 settings (Appendix E). On the contrary, the baseline models use fixed learning rate = 0.01. Apart from that, there are many training params that are used in the training algorithm (Appendix B) but are not mentioned in the paper. They are usually also hard to tune right, including $S$ pre-training steps, $p$ times $F$ update and $t$ times $\Theta$ update per iteration.
  - The paper claims that alternating optimization provides better flexibility than end-to-end training. This is not a well-supported statement. In practice, end-to-end training is much easier to setup, does not require training tricks, and depends less on hyper-params tuning.

**Summary Of The Paper:**

The paper proposes a way to map node features $\boldsymbol{X_i}$, graph structure $\boldsymbol{A}$, and node label $\boldsymbol{Y_i}$ into a shared latent feature space $\boldsymbol{F_i}$. Each node $i$ is now represented by a vector $\boldsymbol{F_i}$, which has the same dimension as one-hot encoded label $Y_i$. $\boldsymbol{F_i}$ is treated as the soft pseudo label of node $i$.

The $X_i \mapsto F_i $ mapping is done by an MLP with learnable $\Theta$ params. The graph structure $\boldsymbol{A}$ is enforced by imposing Laplacian smoothness of $\boldsymbol{F_i}$ against its neighbors $\boldsymbol{F_j}$. The node label $\boldsymbol{Y_i}$ is enforced by otimizing MSE between $\boldsymbol{Y_i}$ and $\boldsymbol{F_i}$ directly.

The paper also proposes an alternative training scheme to learn $\boldsymbol{F}$ and $\Theta$ with various training tricks such as: initial feature diffusion, MLP pre-training that uses only $\boldsymbol{Y_i}$, importance sampling based on $\boldsymbol{F_i}$ entropy, class balancing...

The experiment results show that the proposed methods yield good performance when the label rate is very low.



**Summary Of The Review:**

The paper's novelty is not significantly strong. Most of its empirical positive results are from training tricks that are introduced by Iscen et al. 2019. The paper claims that alternative training is more flexible and more efficient than end-to-end training, which are not well-supported.

---

> ### Author Response · Authors · 2022-11-19
> **Response to Reviewer ogUi - Part 1**
>
> Dear reviewer,
>
> Thanks for the valuable feedback. Here we provide detailed responses to address the concerns in your comments.
>
> Q1: The whole training scheme is very similar to the Iscen et al. 2019 "Label Propagation for Deep Semi-supervised Learning" paper, including most of the important training tricks.
>
> Answer: It is true we use the same Pseudo-label certainty and class balancing trick as Iscen et al. 2019. We never consider these tricks to be our contributions, and have also cited this paper. Actually, the pseudo label selection and re-weighting is two widely used tricks in Pseudo Labeling approaches [1,2,3]. Our contributions mainly focus on two parts:
> * We provide a unified framework from multi-view learning perspective to design numerous GNNs. In section 2.1, we give various options for the three distance metrics.
> * Multiview4GNN has much better computation and memory efficiency than the Persistent propagation GNNs while preserving their performance. From Table 4 in our paper, the training time of Multiview4GNN is slightly more than MLP on both ogbn-arxiv and ogbn-products dataset. Meanwhile, the memory is also less than GCN and APPNP. In addition, our method only need to train an MLP, which is simple to do batch-training, and can significantly reduces the memory cost.
>
> [1] Rizve, Mamshad Nayeem, et al. "In defense of pseudo-labeling: An uncertainty-aware pseudo-label selection framework for semi-supervised learning. ICLR 2021.
>
> [2] Shi, Weiwei, et al. "Transductive semi-supervised deep learning using min-max features." Proceedings of the European Conference on Computer Vision (ECCV). 2018.
>
> [3] Lee, Dong-Hyun. "Pseudo-label: The simple and efficient semi-supervised learning method for deep neural networks." Workshop on challenges in representation learning, ICML. Vol. 3. No. 2. 2013.
>
> Q2: Experiment results only show good improvement on very low label setting (5-60 examples per class). For most of the time, the improvements are not statistically significant.
>
> Answer: For the semi-supervised node classification task, the labeling rate is usually low. For example, the most common setting for Cora, CiteSeer, Pubmed, CS, Physics, Computer and Photo dataset is 20 labels per class for training. As demonstrated in Appendix D, Multiview4GNN outperforms baselines on all of these datasets when using the most common setting. Besides, when the labeling rate is high, the difference between different methods becomes smaller. Even Label Propagation can works well in some datasets, such as Cora, when the labeling rate is high.
>
> Besides, the focus of Multiview4GNN is not to improve the node classification performance. We want provide a new perspective to design GNNs and reduce the time and memory complexity.
>
>
> Q3: Hyper-param search space for the proposed method is much larger than baseline models. The baseline models use fixed learning rate = 0.01.
>
> There are two hyper-parameters, i.e., $\lambda_1$ and $\lambda_2$in our model, and there are a total 25 combinations for the semi-supervised node classification task. For APPNP, there are 11 options for the alpha (from 0 to 1 with step size 0.1). Although there is one more parameters and 2 times more combinations than APPNP, Multiview4GNN is more efficient than APPNP in selecting hyper-paramters. There are two reasons:
>
> * In the hyper-parameters sensitivity ablation study, Figure 4 demonstrates that Multiview4GNN is not very sensitive to these two hyper-parameters at the chosen regions.
>
> * The training time of Multiview4GNN is much less than APPNP. For example, we can find that the training time of APPNP on the ogbn-products dataset is 9 times that of Multiview4GNN from Table 4.
>
> The most commonly used learning rate for training GNNs is 0.01. For multiview4GNN, we also use the same learning rate. In the experiments, our baselines are well-tuned and the results are consistent with the PyTorch-Geometric paper [4].
>
> [4] Fey, Matthias, and Jan Eric Lenssen. "Fast graph representation learning with PyTorch Geometric." arXiv preprint arXiv:1903.02428 (2019).

---

> ### Author Response · Authors · 2022-11-19
> **Response to Reviewer ogUi - Part 2**
>
> Q4: There are many training params that are used in the training algorithm (Appendix B) but are not mentioned in the paper. They are usually also hard to tune right, including  pre-training steps S, $F$ update steps $p$, and $\Theta$ update steps t.
>
> Answer: We did not state the settings of these training parameters clearly in the original paper. Here, we provide detailed explanations: $p$ is fixed to 10 in all the experiments. For Multiview4GNN, we first fixed the training epochs. For example, the training epochs $e$ is set to 1,000 for ogbn-products dataset and 500 for all other datasets. Then, a propagation times k is chosen (usually be 5). Afterwards, we evenly split the training epochs $e$ to $k$ parts. For the first $e/k$ epochs, we pretrain the model, and then do propagation once. For the next $e/k$ epochs we train the $MLP$ and then propagate once, and so on.
>
> Q5: The paper claims that alternating optimization provides better flexibility than end-to-end training. This is not a well-supported statement. In practice, end-to-end training is much easier to setup, does not require training tricks, and depends less on hyper-params tuning.
>
> Answer:  In our paper, we classify the end-to-end training GNNs into two categories, Persistent propagation methods and One-time propagation methods. For the Persistent propagation methods, such as GCN and APPNP, they need to do propagation each epoch. For the One-time propagation methods, such as SGC and SIGN, they only need to propagate once during the training. The flexibility here means that our Multiview4GNN can propagate $k$ times during the training instead of 1 or all, resulting in good performance, with high time and memory efficiency. The downside is that we need to manually select some training parameters, but some simple strategies used in this paper can work well.
>
> Overall, we hope that we have addressed the concerns in your comments, and please kindly let us know if there is any further concern, and we are happy to clarify.

---

### Official Review · Reviewer_YbZy · 2022-10-24

**Confidence:** 4
**Correctness:** 3
**Technical Novelty And Significance:** 3
**Empirical Novelty And Significance:** 2
**Recommendation:** 3

**Clarity, Quality, Novelty And Reproducibility:**

The manuscript is clear and written with good quality. \
The approach is relatively novel but it's limited. \
MULTIVIEW4GNN is relatively straightforward so reproducibility shouldn't be an issue here.

**Strength And Weaknesses:**

**Strengths**
- The paper is well-written and clear.
- The method shows strong performance in very low node label scenarios.
- MULTIVIEW4GNN is memory and time efficient.

**Weaknesses**
- D_A assumes that graphs have high homophily ratios (the smooth assumption). What would the model perform when the graph is heterophily?
- The performance is promising mostly in the scenarios where we have very few labels.
- In my opinion, more rigorous experimentation is needed. The datasets are mostly small, easy to solve, and not diverse.
- Is it mentioned anywhere in the paper what would be the case if there are no initial node features?

**Summary Of The Paper:**

This paper proposes using node features, graph structure, and node labels as three different views and maximizing their agreement.  Therefore, the loss function has three terms and gets optimized with an alternating optimization method.

**Summary Of The Review:**

Most of the points are mentioned in the strength and weaknesses section but in summary, the authors suggest that we can see different types of information as different "views" and try to maximize the agreement between these views with an alternation optimization algorithm. They show that the model is efficient and has good performance, mostly in very low node label scenarios. The datasets in the experiment section are small and not diverse.

---

> ### Author Response · Authors · 2022-11-19
> **Respone to Reviewer YbZy**
>
> Dear reviewer,
>
> Thanks for the valuable feedback. Here we provide detailed responses to address the concerns in your comments.
>
> Q1: The proposed method assumes that graphs have high homophily ratios (the smooth assumption). What would the model perform when the graph is heterophily?
>
> Answer: Our Multiview4GNN is designed for homophily graphs like GCN, APPNP and many popular models. There is no specific design for heterophily graph. For the alternating optimization procedure, one step is to use the feature enhanced label propagation to generate some pseudo labels. Similar to label propagation method, our Multiview4GNN is not suitable for heterophily graphs.
>
> Q2: The performance is promising mostly in the scenarios where we have very few labels.
>
> Answer: It is true that our method works well when the low labeling rate is low and it can have comparable performance with other baselines when labeling rate is high. However, the design of Multiview4GNN is not focus on improving the node classification performance. There are two purposes for Multiview4GNN:
>
> * Provide a unified framework from multi-view learning perspective to design numerous GNNs. In section 2.1, we give various options for the three distance metrics.
>
> * Improve the computation and memory efficiency of GNNs while preserving performance. From Table 4 in our paper, the training time of Multiview4GNN is slightly more than MLP on both ogbn-arxiv and ogbn-products dataset. The training time of APPNP on the ogbn-products dataset is 9 times that of Multiview4GNN.  Meanwhile, the memory cost is also less than GCN and APPNP.
>
> For the semi-supervised node classification task, the labeling rate is usually low. For example, the most common setting for Cora, CiteSeer, Pubmed, CS, Physics, Computer and Photo dataset is 20 labels per class for training. As demonstrated in Appendix D, Multiview4GNN outperforms baselines on all of these datasets when using the most common setting. Besides, when the labeling rate is high, the difference between different methods becomes smaller. Even Label Propagation can works well in some datasets, such as Cora, when the labeling rate is high.
>
> Q3: In my opinion, more rigorous experimentation is needed. The datasets are mostly small, easy to solve, and not diverse.
>
> For the semi-supervised node classification tasks, most widely used datasets are small, and many papers only use 3 or 4 datasets [1, 2]. In our work, we use 3 citation datasets,  2 coauthors datasets, 2 Amazon datasets and 2 OGB datasets for the semi-supervised node classification task. The ogbn-products dataset is a large dataset which includes 2.4 Million nodes and 61 Million edges. Besides, for the inductive node classification task, we also introduce 2 different datasets. The detailed dataset statistics are shown in Appendix C.
>
> [1] Klicpera, Johannes, et al. "Predict then propagate: Graph neural networks meet personalized pagerank." ICLR 2019.
>
> [2] Wu, Felix, et al. "Simplifying graph convolutional networks." International conference on machine learning. PMLR, 2019.
>
>
> Q4: Is it mentioned anywhere in the paper what would be the case if there are no initial node features?
>
> Answer: All the datasets used in this paper contain node features. We use diffusion process to preprocess the features, and then use the new features as inputs to our model. In the ablation study (section 3.4), one variant of our model (Multiview4GNN-w/o-diffusion) use the original feature and the results demonstrate the feature diffusion is not the key component in our method when the label rate is not very low.
>
> Overall, we hope that we have addressed the concerns in your comments, and please kindly let us know if there is any further concern, and we are happy to clarify.

---

### Official Review · Reviewer_aMtC · 2022-10-31

**Confidence:** 4
**Correctness:** 1
**Technical Novelty And Significance:** 2
**Empirical Novelty And Significance:** 2
**Recommendation:** 3

**Clarity, Quality, Novelty And Reproducibility:**

Clarity: The description of the method is clear but the complexity comparison need to be improved.
Novelty: it provides a new perspective but maybe not so useful.

**Strength And Weaknesses:**

The regularization framework  $min_F tr(F^T L F) + ||F-Y||^2$, which is used to derive the graph learning models, is not new and it has been well known for decades. Early works such as manifold ranking [1] already used it; in the era of graph neural networks, as the paper mentioned, some works [2] proposed the unified optimization view of GNNs and also used it; in addition, from the perspective of graph signal processing, there are also some works [3,4] which used it as a special case. These works generally directly obtained the solution expression of F, and then use MLP(X) to take place of F. This paper differs in that it added || MLP(X) – F||^2 as another regularization and unroll the optimization process (the methodology of [4] is somehow similar but used a different objective). It does provide a new perspective and still looks a little bit trivial. The understanding of updating F as propagation and updating \Theta as pseudo label generation is interesting.

 However, the biggest problem is I did not see much advantage of the new formulation and optimization. The analysis of the time complexity if also doubtable.  If Multiview4GNN and APPNP only have dimension $c$ in time complexity because of the feature dimension transformation. Why do not we use a different step for SGC? $XW$ part can be first calculated (so the feature dimension is changed to c) and then do the feature aggregation with \sigma(A^n XW). Let us look back at Multiview4GNN, as other methods contains the computation of backpropagation, why is the backpropagation of the MLP in equation (9) not counted in Multiview4GNN? This MLP generally does not only have one layer so the dimension $d$ cannot be omitted, and the backpropagation also takes $t$ steps.

[1] Zhou et al. Learning with Local and Global Consistency. In NIPS 2003.
[2] Ma et al. A Unified View on Graph Neural Networks as Graph Signal Denoising. In CIKM 2021.
[3] NT, Maehara. Revisiting graph neural networks: All we have is low-pass filters[J]. arXiv preprint arXiv:1905.09550, 2019.
[4] Chen et al. BiGCN: a bi-directional low-pass filtering graph neural network. arXiv preprint arXiv:2101.05519, 2021.



**Summary Of The Paper:**

The paper proposes a new perspective of graph learning by regarding node features, graph structures, and node labels as three different views. Different from the previous GNN models, the paper brings in a new latent variable to be learned and shared across all three views. Through a progressive optimization framework, the paper claims the new framework can obtain better computational and memory efficiency.

**Summary Of The Review:**

In general, the paper has some interesting insights but the novelty is limited. And the contribution especially the analysis of the advantage of the complexity has some problems.

---

> ### Author Response · Authors · 2022-11-19
> **Response to Reviewer aMtC**
>
> Q1: The regularization framework $min_F tr(F^TLF) + ||F-Y||^2$ is not new. What is novelty and advantages of the proposed framework.
>
> A1: The $L =  \lambda_1{||MLP(X)-F||_F^2} + tr(F^\top L F) + \lambda_2 {||F_L-Y_L||_F^2}$  is one instance of our unified framework, and there are numerous designs from this framework as shown in Section 2.1.  If we only consider the first two parts of $L$, i.e., $L_1 = min_F \lambda_1{||MLP(X)-F||_F^2} + {tr(F^\top L F)}$, it is the optimization target for many GNNs[1], such as GCN, GAT and APPNP. The forward process (feature propagation) of these GNNs is to optimize $L_1$, and these GNNs can be used in various tasks, such as node classification, link prediction, and graph classification. However, if we only consider the node classification task, the label information is only used in the MSE loss $L_M= min_F ||F_L-Y_L||_F^2$ (or Cross Entropy loss).  **As a result, the optimization for $L_1$ and optimization for $L_M$ are two separate targets.** The forward process (optimize $L_1$) is fixed once we choose the model architecture, for example, the teleportation parameter $\alpha$ and propagation layers of APPNP. After the fixed optimization for $L_1$, we  optimize $L_M$ using labels. **These two optimizations are independent and there might be some conflicts.** If we only consider the last two parts, i.e.,  $L_2 = min_F {tr(F^\top L F)} + \lambda_2 {||F_L-Y_L||_F^2}$, it is the label propagation. Label Propagation uses the label information in the forward process for classification and have only this one optimization target.
>
> Instead of first designing GNNs from $L_1$, and then applying them to specific tasks, in this paper, we change our perspective by considering the node feature, graph structure, and labels in the node classification task as three views. Then, we can find the proposed multi-view learning framework actually integrate the two optimization targets in GNNs for node classification into a single optimization objective. As a result, the two optimization targets can be concurrently optimized in our framework, instead of first optimizing the $L_1$ loss and then optimizing $L_M$ in GNNs.
>
> Another advantage of our framework is efficiency. Our method can achieve comparable performance but significantly better computation and memory efficiency comparing with Persistent propagation methods, such as GCN and APPNP. From Table 4 in our paper, the training time of Multiview4GNN is slightly more than MLP on both ogbn-arxiv and ogbn-products dataset. The training time of APPNP on the ogbn-products dataset is 9 times that of Multiview4GNN. Meanwhile, the memory cost is also less than GCN and APPNP. In addition, our method only need to train an MLP, which is simple to do batch-training. It can significantly reduces the memory cost.
>
> [1] Yao Ma, Xiaorui Liu, Tong Zhao, Yozen Liu, Jiliang Tang, and Neil Shah. A unified view on
> graph neural networks as graph signal denoising. In Proceedings of the 30th ACM International
> Conference on Information \& Knowledge Management, pp. 1202–1211, 2021.
>
> Q2:  The analysis of the time complexity if also doubtable. Why do not we use a different step for SGC?  $XW$ part can be first calculated (so the feature dimension is changed to c) and then do the feature aggregation with $\sigma(A^n XW)$.
>
> A2: SGC is a One-time propagation method. For SGC, we only need to propagate $X$ once by $A^nX$. And then we can train an MLP multiple epochs using the propagated feature. It is true that we can first calculated $XW$ to reduce the dimension to $c$. However, the $W$ is updated at each epoch during training. As a result, if we calculate $XW$ first for SGC and then do propagation with $A^n XW$, we need to do the feature propagation at each epoch due to the changes of $XW$.
>
> Q3: As other methods contains the computation of backpropagation, why is the backpropagation of the MLP in equation (9) not counted in Multiview4GNN? This MLP generally does not only have one layer so the dimension $d$ cannot be omitted, and the backpropagation also takes $t$ steps.
>
> A3: For Multiview4GNN, it also need the backpropagation for MLP. In our time complexity analysis, we omit the forward and backward of MLP layers for all methods including GCN, SGC, and APPNP. We only analyze the cost of forward and backward for the propagation layers. The reason is as follows: We assume these method have the same layers of MLP, and then the cost of forward and backward of MLP in these methods is the same. By omitting the cost of MLP, we can clearly show the difference between our method with Persistent propagation methods and One-time propagation methods.
>
> Overall, we hope that we have addressed the concerns in your comments, and please kindly let us know if there is any further concern, and we are happy to clarify.

---

### Decision · Program_Chairs · 2023-01-20

**Decision:**

Reject

**Justification For Why Not Higher Score:**

The 2 main issues were the novelty and need for clearer comparison to related work, and the need for stronger evidence of the benefits of the proposed framework, as mentioned in my review.

**Justification For Why Not Lower Score:**

N/A

**Metareview: Summary, Strengths And Weaknesses:**

The paper formulates the graph learning task from a multi-view perspective, where the node features $\mathbf{X}$, graph structure $\mathbf{A}$, and node labels $\mathbf{Y}$ are regarded as three views. Latent variables $\mathbf{F}$ and node feature encoder parameters $\mathbf{\Theta}$ are learned in an alternating scheme (allowing for a lazy propagation approach) minimizing three loss components corresponding to the three views.

In general, reviewers agreed that the paper is generally well-written, and offers interesting insight in its perspective of graph learning, and in its training scheme. However, there are a number of key issues raised by the reviewers, particularly:

- Novelty and comparison to related work: reviewers note the similarity to existing work, particularly [1], as well as [2,3]. While the authors note some differences such as the modified optimization approach in this paper, it would still be better to include a clearer description of the relationship between the current and the closely related works, particularly [1] which is only briefly mentioned in the paper.

- Evidence for benefits of proposed framework: regarding experimental performance, reviewers noted that the performance benefits are only relatively clear on low label rates, but not on high label rates. The authors note in their response that the method also provides improvements in computational and memory efficiency; however, one issue is that the (time / memory) efficiency comparison is only clearly favorable for Multiview4GNN-k (but not Full), while the accuracy performance of Multiview4GNN-k is not shown, making the benefits less clear.

- Other issues: reviewers noted that the proposed approach is not suitable for heterophilic graphs (as agreed by the authors), and significantly increases the amount of hyperparameter tuning (due to $\lambda_1, \lambda_2$ and potentially some other hyperparameters). The authors provide further details on the tuning procedure in their response, but there are still some slightly unclear parts (how $e$ and $k$ are chosen).

In the end, PCs and AC agree that while the work has intriguing ideas, due to the above issues, the work is not yet ready for publication at ICLR. The reviews offer a number of helpful suggestions for improvement, so I encourage the authors to continue improving the paper based on the reviews for future submissions.


[1] Iscen et al., Label Propagation for Deep Semi-supervised Learning, CVPR 2019.
[2] Zhou et al. Learning with Local and Global Consistency. NIPS 2003.
[3] Ma et al. A Unified View on Graph Neural Networks as Graph Signal Denoising. In CIKM 2021.